# A predictive model of joint dynamics and ground reaction force using only leg length, body mass, and walking cadence

Huan Zhao [1,2,3], Guowu Wei[4], Junxiao Xie[2], Anmin Liu[5], Qiumin Qu[1], Junyi Cao [2*], Ziyun Ding [6], Wei-Hsin Liao[3]

**1** Department of Neurology, The First Affiliated Hospital of Xi'an Jiaotong University, Xi'an, P.R. China, **2** School of Mechanical Engineering, Xi'an Jiaotong University, Xi'an, P.R. China, **3** Department of Mechanical and Automation Engineering, The Chinese University of Hong Kong, Shatin, N.T., Hong Kong, China, **4** School of Science, Engineering and Environment, University of Salford, Salford, United Kingdom, **5** School of Health and Society, University of Salford, Salford, United Kingdom, **6** School of Engineering, University of Birmingham, Birmingham, United Kingdom

* caojy@mail.xjtu.edu.cn

## Abstract

Reconstructing premorbid gait patterns is critical for developing personalized rehabilitation strategies and assistive devices for patients with movement disorders. To achieve this aim, a predictive model is developed to estimate the walking dynamic features with individual parameters without requiring complex gait tests. First, an empirical kinematic model predicting the joint angle on the basis of leg length and walking cadence is derived. Consequently, dynamic models for the single support phase and double support phase are established, and a linear transformation strategy is proposed in the double support phase for optimization. Using inverse dynamic approaches, the model can ultimately predict the joint angle, joint moment, and ground reaction force across the entire gait cycle using only leg length, body mass, and walking cadence. The dynamic parameters predicted with the model are compared with experimental data for validation, and the results demonstrate the effectiveness of the proposed model.

## 1. Introduction

In neurorehabilitation therapies, helping patients with gait disorders recover gait patterns as physiologically normal as possible is one of the main focuses [1]. For more effective performance, a locomotion-assisting system such as orthoses and exoskeleton devices that can predict the healthy gait pattern of each user is necessary [2,3]. Additionally, many assistants have been proposed as walking aids for patients with movement disorders [4], such as Parkinson's disease [5] and stroke [6], which also require the premorbid walking dynamics of each patient individually.

**Data availability statement:** Our data is publicly posted by the following URL: https://github.com/zhaoxiaohuan13/predictive-model-of-joint-dynamics-and-ground-reaction-force.

**Funding:** This work was supported by the China Postdoctoral Science Foundation [Grant No. GZB20240603] (received by Huan Zhao), the National Key Research and Development Program of China [Grant No. 2021YFE0203400] (received by Junyi Cao), China Scholarship Council [Grant No. 202006280395] (received by Huan Zhao), the Innovation and Technology Commission under the Mainland-Hong Kong Joint Funding Scheme (MHKJFS), the Hong Kong Special Administrative Region, China [Project No. MHP/043/20] (received by Wei-Hsin Liao), Science, Technology, and Innovation Commission of Shenzhen Municipality, China [Grant No. SGDX20220530111005036] (received by Wei-Hsin Liao), and The Chinese University of Hong Kong [Project ID: 3130174, 4055254] (received by Huan Zhao).

**Competing interests:** The authors have declared that no competing interests exist.

A normal gait is fundamental for a healthy and active lifestyle. Modeling of human walking has been investigated for several decades [7,8]. A damped bipedal inverted pendulum model was verified by 14 subjects walking on a treadmill [9]. An extended bipedal spring-loaded inverted pendulum model was combined with a force-modulated compliant hip and velocity-based leg adjustment to generate gait [10]. A bipedal spring–mass model was developed to qualitatively predict the motion of the center of mass during walking. The model used input parameters such as the angle of attack, vertical position, and forward speed of the center of mass, which were extracted from experimental data [11]. The bipedal model is adjusted to fit the vertical ground reaction force (GRF) measured experimentally [12].

Traditional inverted pendulum models of human walking neglect the double support phase and thus cannot produce a double-hump GRF; to describe the GRF accurately, a double support phase is considered [13]. However, the motion equations require input parameters that must be obtained experimentally, including leg length, leg stiffness, leg damping, angle of attack, and initial conditions (e.g., displacement or velocity) [14].

Lower limb joint kinematics and kinetics exhibit bidirectional dynamic coupling during walking [15]. Conventionally, the hip, knee, and ankle moments are estimated by applying inverse dynamic methods with kinematic data of body movement and GRFs [16,17]. The acquisition of kinematic data such as movement trajectories requires high-accuracy equipment such as a motion capture system [18]. The measurement of the GRF and corresponding center of pressure requires a force plate [19].

The relationship between the joint moment and angular velocity during normal human walking can be effectively captured via a single linear controller [20]. To model this relationship, researchers have developed a unified predictive kinematic model comprising Bezier polynomials to represent the trend and Fourier series to capture the periodic component [21]. Owing to the dependence on specialized equipment and technical expertise [18], machine learning methods [22] such as recurrent neural networks with long short-term memory have been employed to estimate the knee abduction moment. However, these methods still require the use of wearable sensors [23]. In addition, machine learning models lack interpretability, which is a crucial factor in many applications. To address this issue, it is essential to establish a predictive model that does not rely on complex equipment.

A simple walking model may result in discontinuities in dynamic parameters such as speed, plantar force, and joint moments during step–to–step transitions. An accurate double support model minimizes discontinuities during step–to–step transitions, promoting symmetrical movement of the lower limbs in both kinematics and dynamics. One of the major challenges in modeling the double support phase is how to appropriately distribute plantar pressure to the ground contact points of both feet. Ren et al. proposed a smoothing assumption [24,25], i.e., linear transformation, to distribute the GRF, specifically by calculating the ratio of the sum of the measured vertical GRF of the left and right feet to the vertical GRF as the distribution coefficient of the vertical GRF and by using the coefficient of the measured horizontal force

to the vertical pressure as the prediction coefficient of the horizontal force. This process assumes that the motion of the left and right feet is symmetrical in the double support phase. Seung et al. addressed the hyperstatic structure problem by using traditional Newton mechanics for the single support phase and an artificial neural network model for the double support phase [26]. This study proposes a linear allocation strategy for body mass during the double support phase to optimize the dynamic analysis of the double support phase and ultimately achieve smooth transitions from step to step.

The main objective of this study is to develop a gait dynamic model capable of predicting kinematics (joint angles) and kinetics (GRF and joint moments) continuously without complex equipment. The model utilizes easily measurable parameters, including leg length, body mass, and walking cadence. The unique contribution of this study is twofold. First, we propose a symmetrical predictive gait dynamic model consisting of an empirical model for joint angle prediction and a kinetic model for GRF and joint moment predictions across the entire gait cycle. Specifically, the novel kinetic model consists of two segments in the double support phase and three segments in the single support phase. Second, we fully validated the model by comparing its predictions with gait data collected from healthy adults via instrumented measurements.

## 2. Methods

A dynamic gait model in the sagittal plane of two symmetrical legs across the entire gait cycle is proposed. The kinematic model is an empirical model with the inputs of leg length and walking cadence. The kinetic model has two segments for the double support phase and three segments for single support. The empirical kinematics model of the joint angle acted as input into the proposed dynamic model for the GRF and joint moment estimations. Consequently, the GRF and moments of the hip, knee, and ankle can be predicted with only the leg length, body mass, and walking cadence. The proposed model was ultimately validated against experimentally measured GRFs, joint angles derived through OpenSim's inverse kinematics pipeline, and joint moments computed via OpenSim's integrated inverse dynamics solution.

### 2.1. Gait data acquisition

The study was approved by the Institutional Review Board of Xi'an Jiaotong University (protocol code 2020-1330). The recruitment period for participants was from November 1, 2020, to June 30, 2021. All participants provided written informed consent. Fourteen healthy adult individuals participated (5 females and 9 males; mean age: $25 \pm 2$ years; mean height: $167.9 \pm 10.1$ cm; mean body mass: $58.7 \pm 10.3$ kg). The participants had not experienced any lower-limb musculoskeletal injuries for at least three years before testing.

Motion trajectories were captured at a rate of 100 Hz via a 12-camera optical system (Vicon MX, OML, UK). GRF data were collected at 1000 Hz from force plates (AMTI, 40060, Advanced Mechanical Technology, Inc., Watertown, MA, USA) [27]. The subjects walked at their self-selected pace while outfitted with 16 retroreflective markers, as described in our previous work [28]. Body mass and leg length were measured for each subject before the experiment. All the subjects walked barefoot at their preferred cadence on a 7-meter track equipped with three embedded force plates.

The subjects were divided into two groups: 10 subjects were employed to obtain the empirical parameters in the predictive model of the joint angle, and the remaining 4 subjects acted as testing subjects to validate the proposed predictive dynamic model.

### 2.2. Predictive joint angle modeling

The Fourier series-based empirical formulas for the motion of the knee and hip joints from 10 subjects are detailed in [28] as follows:

$$\phi_L = B_0 l + \sum_{j=1}^{n} B_j l \sin(2\pi j f t + \varphi_j) \tag{1}$$

where $\phi_L$ refers to the joint angle of the left hip or knee, and the unit of which is "Radian", $l$ is the leg length, which is the measured distance from the anterior superior iliac spine (ASIS) to the medial malleolus via the tape measure; its unit in this equation is "m". This distance is different from the effective leg length, which is the distance between the center of mass and the contact point [29]. $f$ corresponds to the walking cadence, and it is the number of strides in one second, it can be measured by the assessor using a stopwatch to record a 60-second interval, counting the number of steps (initial contacts of the same foot) within that period, the cadence can also derive the gait cycle $T$ as $T = 1/f$. $B_j$ denotes the empirical coefficient of the amplitude and is expressed in units of Rad/m, $j$ represents the serial number of harmonic orders, $n$ denotes the total number of harmonic orders, $t$ is the time, and $\varphi_j$ refers to the empirical parameter of the initial phase.

$$\phi_R = B_0 l + \sum_{j=1}^{n} B_j l \sin(2\pi jft + \varphi_j + j\pi) \tag{2}$$

where $\phi_R$ is the predicted joint angle of the right hip or knee.

Since $n$ is the total number of harmonics, it was determined based on a preliminary analysis which aimed to capture the essential dynamics of the signal while avoiding overfitting as illustrated in our previous research [28]. For thigh and knee joints, the first three harmonics were selected, while for ankle joint, the first four harmonics were selected, which provided an optimal balance between reconstruction accuracy and model complexity. In addition, the coefficients of the Fourier series "$B_j$" were tuned by linear least-squares regression to minimize the error between the approximated signal and the original input data as demonstrated in [28].

### 2.3. Predictive gait kinetics modeling

The hip and knee primarily extend and flex the thigh and shank during lower limb movement, enabling substantial overall body motion. The simplified bipedal walking model proposed in this study ignores the function of the foot, focusing on the interaction between the two lower limbs. The input data from multiple gait cycles were time-normalized and averaged to produce a single, representative gait cycle for each participant. In addition, for a simpler model, the standing leg is simplified as a rigid rod fixed to the ground, and the swinging leg is simplified as two articulated rigid links, as shown in Fig 1. $M_h$ represents the moment at hip for the supporting leg, $M_{h2}$ represents the moment at hip for the swing leg, $M_a$ represents the moment at ankle, and $M_k$ represents the moment at knee.

1) Bipedal single support phase dynamic model

A simplified human lower limb structure, represented by a rigid body model, is established with the foot in the stance as the system's origin, as depicted in Fig 2. The standing leg is a rigid rod connected to the ground, with the upper body mass attached to the end of the standing leg in the form of a concentrated mass point. On the contralateral lower limb, i.e., the swinging leg, the thigh is articulated with the mass point to form the hip joint, and the shank is articulated with the thigh, representing the knee joint of the swinging leg. During the walking process, a moment is generated at the ankle joint of the standing leg, and a moment is generated at the hip and knee joints of the swinging leg.

In the Cartesian coordinate system with the foot in the stance phase as the origin, the Cartesian absolute angles are used to represent the centroid position and velocity of the standing leg.

$$\begin{bmatrix} x_i \\ y_i \end{bmatrix} = \begin{bmatrix} \sum_{i=0}^{i-1} l_{i-1}\sin\theta_{i-1} \\ \sum_{i=0}^{i-1} l_{i-1}\cos\theta_{i-1} \end{bmatrix} + \begin{bmatrix} p_i\sin\theta_i \\ p_i\cos\theta_i \end{bmatrix} \tag{3}$$

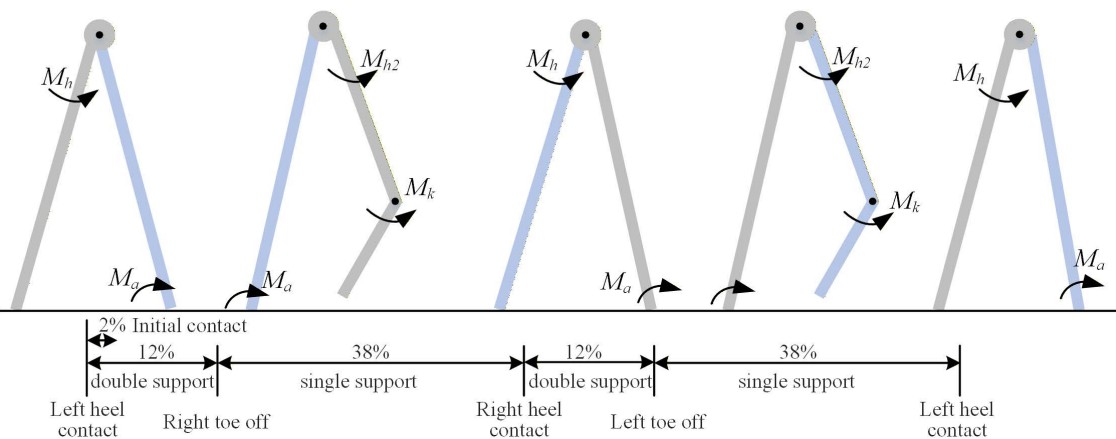

**Fig 1. The proposed kinetic model across the entire gait cycle.** The blue rod represents the left leg, and the gray rod represents the right leg during walking. The gray circle represent the trunk with the pelvis. The gait cycle is initiated at left heel contact (0%) and is normalized to 100%. The diagram depicts the double support phases and single support phases. The first double support phase (0-12%): the period when both the feet is in contact with the ground (for example, the right leg is the trailing leg); The first single support phase (13-50%): the period when only one foot is on the ground (the right leg is swing, and left foot contact the ground); The second double support phase (51-62%): the period when both the feet is in contact with the ground (the right leg is the dominant leg). The first single support phase (63-100%): The period when only one foot is on the ground (the right leg is supporting, and right foot contact the ground).

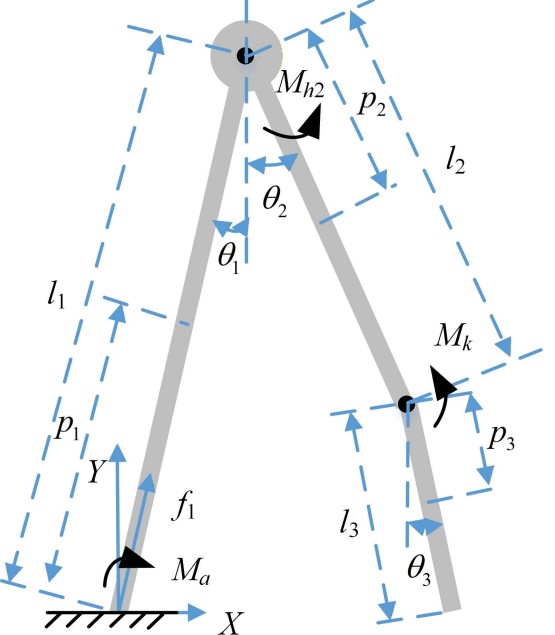

**Fig 2. Three-link dynamic model of a single support phase.**

where $x_i$ and $y_i$ represent the horizontal and vertical displacement of the centroid of the $i$ th segment, $l_{i-1}$ represents the length of the ($i$-1) th segment, $p_i$ represents the distance between the centroid and the starting point of the $i$ th segment, and $\theta_i$ represents the angle between the $i$ th segment and the vertical direction, $i$ is ranged from [1–3] for single support

phase, when $i$ is 1, it represents the standing leg, when $i$ equals to 2 it represents the swing thigh, and when $i$ is 3 it is the swing shank,.

$$\begin{bmatrix} \dot{x}_i \\ \dot{y}_i \end{bmatrix} = \begin{bmatrix} \sum_{i=0}^{i-1} l_{i-1}\cos\theta_{i-1} \\ \sum_{i=0}^{i-1} l_{i-1}\sin\theta_{i-1} \end{bmatrix} \dot{\theta}_{i-1} + \begin{bmatrix} p_i\cos\theta_i \\ p_i\sin\theta_i \end{bmatrix} \dot{\theta}_i$$

(4)

where $\dot{x}_i$ and $\dot{y}_i$ denote the horizontal and vertical velocities of the centroid of the $i$ th segment, respectively, and $\dot{\theta}_1$ refers to the angular velocity of the $i$ th segment.

2)  Bipedal double support phase dynamic model

In the double support phase, the angles of all body segments are not entirely independent because the model forms a closed-loop structure. In this study, we assume that during the double support phase, the thigh and shank act as a single entity, forming a rigid body, as illustrated in Fig 3. The "leading leg" is the leg positioned in front of the body's trunk. The "trailing leg" (or back-swinging leg) is the leg that lags behind the body's trunk.

During double support phase, to distinguish the differences from the single support phase, the corresponding joint angles are added with a subscript "$d$" after 1 and 2. $\theta_{1d}$ represents the angle between the trailing leg and vertical direction, while $\theta_{2d}$ denotes the angle between the leading leg and vertical direction, and $p_{2d}$ corresponds to the distance between the centroid of the leading leg and the hip.

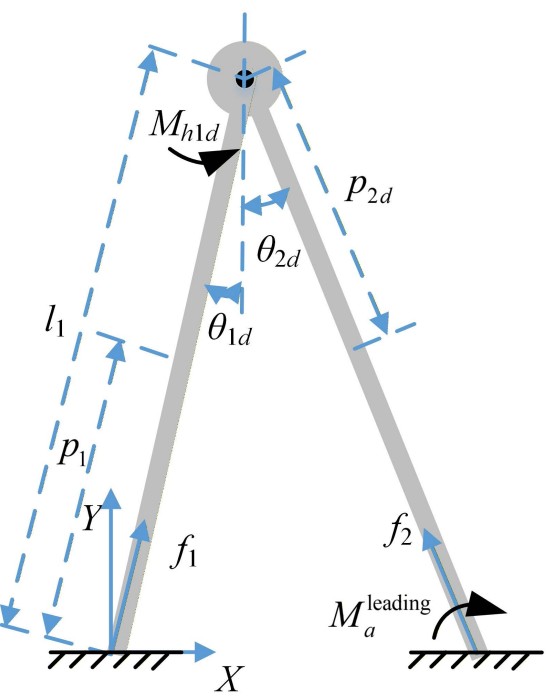

**Fig 3. Two-link dynamic model of the double support phase.** The limbs are assumed to be symmetrical.

## 2.4. Prediction of the GRF

During walking, the foot experiences vertical and anterior–posterior GRFs due to the body's weight and acceleration. The horizontal force is induced from the friction between the foot and the ground, as well as resistance and inertia. It appears in both the lateral-medial direction and the anterior-posterior direction. Since this study focuses on the sagittal plane, the horizontal force mentioned later denotes the force in the anterior–posterior direction, with the mediolateral force being ignored in this study.

1) GRF during the single support phase

The vertical GRF can be calculated via Newton's second law of motion as follows:

$$GRF_y - 2(m_t + m_s)g - mg = (m_t + m_s)\ddot{y}_s + m\ddot{y}_c + m_t\ddot{y}_{wt} + m_s\ddot{y}_{ws} \tag{5}$$

where $m_t$ denotes the mass of the thigh; $m_s$ refers to the mass of the shank; and $m$ is the sum of the masses of the trunk, head, and upper limbs. They are determined by the subject's body mass and the relative mass coefficient, as specified in the "National Standard of the People's Republic of China: Inertial Parameters of Adult Human Body (GB/T17245-2004)". $\ddot{y}_s$ refers to the vertical acceleration of the stance leg, and $\ddot{y}_{wt}$ and $\ddot{y}_{ws}$ correspond to the vertical accelerations of the swing thigh and shank, respectively.

The anterior–posterior GRF was derived as follows:

$$GRF_x = (m_t + m_s)\ddot{x}_s + m\ddot{x}_c + m_t\ddot{x}_{wt} + m_s\ddot{x}_{ws} \tag{6}$$

where $\ddot{x}_s$ refers to the horizontal acceleration of the stance leg, $\ddot{x}_{wt}$ denotes the horizontal acceleration of the swing thigh, and $\ddot{x}_{ws}$ is the horizontal acceleration of the swing shank.

2) GRF during the double support phase

The vertical GRF was calculated as follows:

$$GRF_y - 2(m_t + m_s)g - mg = (m_t + m_s)\ddot{y}_{s1} + m\ddot{y}_c + (m_t + m_s)\ddot{y}_{s2} \tag{7}$$

where $\ddot{y}_{s1}$ and $\ddot{y}_{s2}$ refer to the vertical acceleration of the trailing and leading legs, respectively.

The anterior–posterior GRF was calculated as follows:

$$GRF_x = (m_t + m_s)\ddot{x}_{s1} + m\ddot{x}_c + (m_t + m_s)\ddot{x}_{s2} \tag{8}$$

where $\ddot{x}_{s1}$ refers to the horizontal acceleration of the trailing leg and where $\ddot{x}_{s2}$ refers to the horizontal acceleration of the leading leg.

However, the GRF calculated during the double support phase is the sum of the forces from both the left and the right feet. Existing methodologies have commonly adopted a standardized approach to quantify bilateral loading by determining the left and right foot GRFs through a distribution coefficient derived from the ratio of the summed measured GRF of both limbs to the total GRF [24,25]. To obtain the GRF of each foot more accurately, it is necessary to propose a novel physical model for the double support phase.

## 2.5. Novel double-support phase modeling

The traditional double support phase derives the GRF with the experimental ratio [25]. Our study assumes that during walking, the body mass is linearly transferred from the trailing leg to the leading leg from the moment of landing to the end

of the double support phase, and the body mass borne by the trailing leg linearly decreases from the original total mass to zero. On the basis of this assumption, the refined calculation for the vertical GRF under each foot in the double support phase can be derived.

$$\begin{cases} GRF_y^{s1} = g(2(m_t+m_s)+m) - g(2(m_t+m_s)+m)(t-t_{\text{heel strike}}^{s2})/T_{DT}+m\ddot{y}_c/2+(m_t+m_s)\ddot{y}_{s1} \\ GRF_y^{s2} = g(2(m_t+m_s)+m)(t-t_{\text{heel strike}}^{s2})/T_{DT}+m\ddot{y}_c/2+(m_t+m_s)\ddot{y}_{s2} \end{cases}$$

(9)

where $GRF_y^{s1}$ denotes the vertical ground reaction of the trailing leg; $GRF_y^{s2}$ is the vertical ground reaction of the leading leg; $t$ refers to the time series in the double support phase $\{t_{\text{heel strike}}^{s2}, \cdots, t_{\text{toe off}}^{s1}\}$; $t_{\text{heel strike}}^{s2}$ represents the heel strike time point of the leading leg; $t_{\text{toe off}}^{s1}$ represents the toe-off time point of the trailing leg; $T_{DT}$ represents the time of the double support phase; and $T_{DT}=t_{\text{toe off}}^{s1} - t_{\text{heel strike}}^{s2}$.

The calculation for the anterior–posterior GRF in the forward direction can be derived as

$$\begin{cases} GRF_x^{s1} = (m-m(t-t_{\text{heel strike}}^{s2})/T_{DT})\ddot{x}_c/2+(m_t+m_s)(t-t_{\text{heel strike}}^{s2})/T_{DT}\ddot{x}_{s1} \\ GRF_x^{s2} = (m+m(t-t_{\text{heel strike}}^{s2})/T_{DT})\ddot{x}_c/2+(m_t+m_s)(t-t_{\text{heel strike}}^{s2})/T_{DT}\ddot{x}_{s2} \end{cases}$$

(10)

where $GRF_x^{s1}$ is the GRF in the forward direction of the trailing leg and where $GRF_x^{s2}$ corresponds to the GRF in the forward direction of the leading leg.

## 2.6. Prediction of joint moments

The Lagrangian method was used to calculate the joint moments of the lower limbs.

$$\frac{d}{dt}\left(\frac{\partial L}{\partial \dot{q}_i}\right) - \frac{\partial L}{\partial q_i}=Q_i$$

(11)

where $Q_i$ denotes the generalized forces derived from the virtual work ($\delta W$), $\delta W=\sum_i Q_i\delta q_i$, and $\delta q_i$ is the change in the state vector $q_i$, which can be either joint angles $\phi_i$ or the angles between body segments and the vertical direction $\theta_i$.

$$\delta W=\sum_i -M_i\delta\phi_i = \sum_i M_i(\theta_i-\theta_{i-1}) = \sum_i (M_{i+1}-M_i)\theta_i$$

(12)

where $M_i$ is the $i$-th joint moment.

The angles in the dynamic model, as shown in Figs 1 and 2, are the angles between body segments and the vertical axis of the world coordinate system, denoted as $\theta_i$, when $i=1$, $\theta_i$ is the angle of the left thigh to vertical axis $\theta_{hip}$, when $i=2$, $\theta_i$ is the angle of the right thigh to vertical axis $\theta_{hip}$, when $i=3$, $\theta_i$ is the angle of the right shank to vertical axis $\theta_{knee}$, and it can be derived by the joint angle Φ, and $\theta_{knee}=\theta_{hip}-\phi_{knee}$. As assumed in Section 2, the angle of the pelvis in the sagittal plane is zero during walking; thus, $\theta_{hip}=\phi_{hip}$.

$$\frac{d}{dt}\left(\frac{\partial L}{\partial \dot{\theta}_i}\right) - \frac{\partial L}{\partial \theta_i}=M_{i+1} - M_i$$

(13)

where $L=E_{\text{kinetic}} - E_{\text{position}}$, $E_{\text{kinetic}}$ is the kinetic energy of the model, and $E_{\text{position}}$ is the position energy of the model, the details of which are displayed in the supplemental materials.

Taking the origin as the zero potential energy point, the moment of the ankle (the moment acting on the end of the leg that contacts the ground), hip, and knee in the single support phase and double support phase can be obtained via Eq. (17) and Eq. (18), respectively.

$$
\begin{cases}
M_k = \dfrac{\mathrm{d}}{\mathrm{d}t}\dfrac{\partial L}{\partial \dot{\theta}_3} - \dfrac{\partial L}{\partial \theta_3} \\[6pt]
M_k - M_{h2} = \dfrac{d}{dt}\dfrac{\partial L}{\partial \dot{\theta}_2} - \dfrac{\partial L}{\partial \theta_2} \\[6pt]
M_{h2} - M_a = \dfrac{\mathrm{d}}{\mathrm{d}t}\dfrac{\partial L}{\partial \dot{\theta}_1} - \dfrac{\partial L}{\partial \theta_1}
\end{cases}
\tag{14}
$$

$$
\begin{cases}
M_a^{\text{leading}} = \dfrac{\mathrm{d}}{\mathrm{d}t}\dfrac{\partial L}{\partial \dot{\theta}_{2d}} - \dfrac{\partial L}{\partial \theta_{2d}} \\[6pt]
M_a^{\text{leading}} - M_h^{\text{trailing}} = \dfrac{d}{dt}\dfrac{\partial L}{\partial \dot{\theta}_{1d}} - \dfrac{\partial L}{\partial \theta_{1d}}
\end{cases}
\tag{15}
$$

In the established model, since the double support phase directly integrates the knee joint with the lower leg and the standing leg of the single support phase also integrates the knee joint with the thigh and shank, the knee joint moments of the trailing leg in both the double support phase and the single support phase are calculated together with the hip and ankle joint moments of the same lower limb. The knee moment in the standing phase for both the single support phase and the double support phase can be derived as:

$$
M_k^{\text{stand}} = M_a^{\text{stand}}\frac{l_3}{l_1} + M_h^{\text{stand}}\frac{l_2}{l_1}
\tag{16}
$$

$M_k^{\text{stand}}$ superscript "stand" denotes that the lower limb is in the stance phase, the subscript refers to the joint, $k$ $M_k^{\text{stand}}$ corresponds to the knee joint, $a$ $M_a^{\text{stand}}$ corresponds to the ankle joint, and $M_h^{\text{stand}}$ corresponds to the hip joint.

## 2.7. Prediction of kinetic parameters across the entire gait cycle

On the basis of the assumption of complete symmetry between the two lower limbs, the analysis of a full gait cycle can be conducted using half a gait cycle, which consists of a double support phase followed by an immediately adjacent single support phase. Substitute the predicted formulas for joint angles of the left and right lower limbs, expressed in terms of individual walking cadence and leg length (obtained in Section 2 Eq. (1) and Eq. (2)) into Eq. (18) and Eq. (19), and the double support phase accounts for 12% of the gait cycle; the hip and ankle joint moments during the double support phase are derived. Similarly, by substituting the joint angle predicted by Eq. (1) and Eq. (2) into Eq. (17) and Eq. (19) and considering the single support phase as 38% of the gait cycle, the hip, knee, and ankle joint moments during the single support phase can be calculated.

## 2.8. Data analysis

To evaluate the performance of the unified prediction model, the root mean square error (RMSE) was calculated as

$$
\text{RMSE} = \sqrt{\frac{1}{N}\sum_{t=1}^{N}(y_t - \hat{y}_t)^2},
\tag{17}
$$

where $y_t$ is the predicted value at time '$t$', $\hat{y}_t$ is the measured value at time '$t$', and $N$ is the total number of time points. Furthermore, the relative RMSE (rRMSE) was derived as:

$$
\text{rRMSE} = \frac{\text{RMSE}}{(\max(\hat{y}_t) - \min(\hat{y}_t))},
\tag{18}
$$

where $\max(\hat{y}_t)$ is the maximum value of the measured value and where $\min(\hat{y}_t)$ is the minimum of the measured value.

The Pearson correlation coefficient was calculated as follows:

$$\rho = \frac{\sum\limits_{t=1}^{N}(y_t-\bar{y})(\hat{y}_t-\bar{\hat{y}})}{\sigma(y)\sigma(\hat{y})} = \frac{\sum\limits_{t=1}^{N}(y_t-\bar{y})(\hat{y}_t-\bar{\hat{y}})}{\sqrt{\sum\limits_{t=1}^{N}(y_t-\bar{y})^2}\sqrt{\sum\limits_{t=1}^{N}(\hat{y}_t-\bar{\hat{y}})^2}},$$

(19)

where $\rho$ represents the correlation coefficient of the trajectory predicted by the model and the trajectory measured experimentally, $\bar{y}$ represents the mean value of the trajectory predicted by the model, $\bar{\hat{y}}$ represents the mean value of the trajectory measured experimentally, $\sigma(y)$ represents the standard deviation of the trajectory predicted by the model, and $\sigma(\hat{y})$ represents the standard deviation of the trajectory measured experimentally.

Statistical comparisons between predicted and experimentally collected signals were performed using one-dimensional Statistical Parametric Mapping (SPM) via the open-source SPM1D toolbox for MATLAB (https://spm1d.org/index.html) at a significance level of $\alpha = 0.05$. Clusters where SPM{t} exceeded the corresponding critical threshold (t*) were considered statistically significant, and that indicates the predicting model is invalid.

## 3. Results

### 3.1. Predicted joint kinematics

For the joint angles of the hip and knee joints in the sagittal plane, the empirical parameters summarized from the 10 subjects are presented in Table 1.

With the empirical parameters in Table 1, the predictive mathematical model for the joint angle was completed. Compared with the movement data collected with the VICON system, the joint angles of the 4 test subjects predicted by empirical formulas using only leg length and walking cadence were compared. Since the prediction results of the model are in radians, they are converted to degrees for direct comparison with experimentally obtained data, which are in degrees. The predicted angles are compared with the experimental data of 8 trials for one subject randomly selected from the testing subjects, as shown in Fig 4.

The results demonstrate a strong consistency between the individual joint angle changes predicted by the unified parameterized model and those measured in the experiments. The rRMSE values and correlation coefficients of the hip joint angles, knee joint angles, and experimentally measured joint angles for the 4 test subjects based on leg length and walking cadence are shown in Table 2.

Table 2 clearly shows that the average rRMSE values for the 4 test subjects are less than 10%, with average correlation coefficients ranging from 0.98--0.99. These results indicate the efficiency of the predictive model.

### 3.2. The predicted GRF

The GRFs predicted by the model and measured by the force plate were normalized to the body weight of the subject and then compared, as shown in Fig 5.

**Table 1. Empirical parameters in the predictive model of the hip and knee in the sagittal plane.**

| Parameter | $B_0$ | $B_1$ | $B_2$ | $B_3$ | $\varphi_1$ | $\varphi_2$ | $\varphi_3$ |
|---|---|---|---|---|---|---|---|
| Hip | 0.086 | −0.316 | −0.067 | 0.026 | −1.105 | 1.433 | 0.187 |
| Knee | 0.468 | 0.465 | 0.311 | −0.093 | 0.244 | −0.990 | 0.266 |

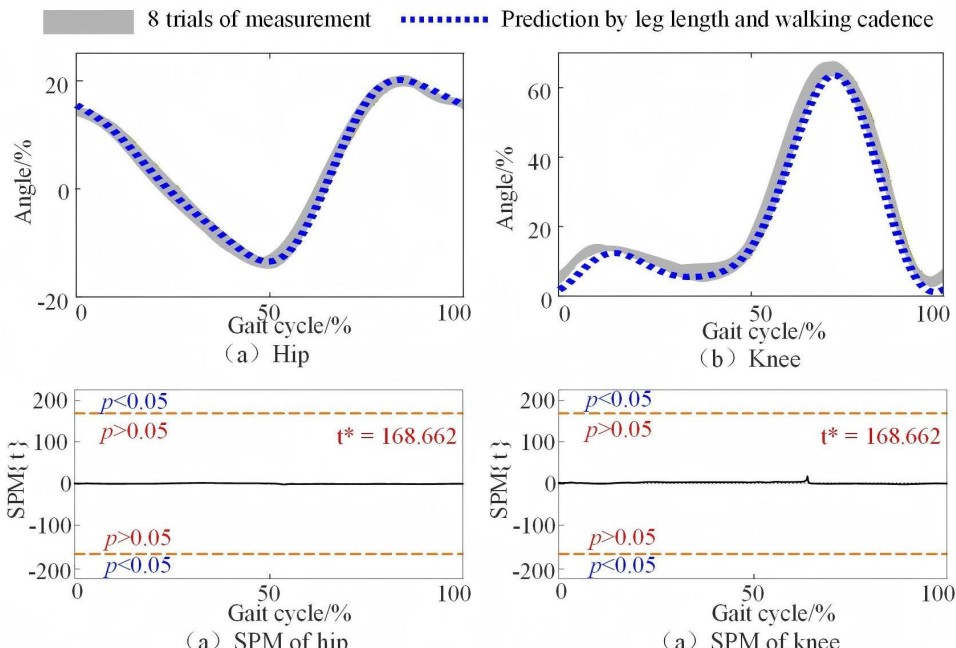

Fig 4. Measurement of the hip angle and knee angle in the sagittal plane of one subject. The horizontal axis of the gait cycle is determined on the basis of the left foot. The orange red dashed line indicates the critical thresholds (t* = 168.662, α = 0.05). Regions of the gait cycle for which SPM {t} exceeded the critical threshold are considered as the predicting model is invalid.

Table 2. Error evaluation of the joint angle prediction model.

| Tester | Hip | | Knee | |
|---|---|---|---|---|
| | rRMSE/% | ρ | rRMSE/% | ρ |
| 1 | 6.34 | 0.99 | 9.36 | 0.99 |
| 2 | 6.27 | 0.98 | 7.48 | 0.98 |
| 3 | 6.42 | 0.98 | 13.01 | 0.96 |
| 4 | 10.48 | 0.99 | 9.52 | 0.99 |
| mean | 7.38 | 0.99 | 9.84 | 0.98 |

The overall trend of the vertical and anterior–posterior GRFs predicted by walking cadence, body mass, and leg length is similar to the force change curve of the walking process obtained from the force plate experimentally. Compared with the vertical GRF, the anterior–posterior GRF is smaller, and the model error is relatively larger.

The predicted data were in accordance with the experimental measurements, with some errors. The rRMSE and correlation coefficient were calculated from the mean value of the 8 walking trials, and the results from the 4 testing subjects are shown in Table 3. The rRMSE values were no greater than 11.83%, and the correlation coefficient was no less than 0.93.

### 3.3. Predicted joint moments

The joint moments predicted by the model with body mass, leg length, and walking cadence were compared to the experimental joint moments of 8 walking trials of the same subject, as shown in Fig 6.

The estimated ankle joint moments based on individual parameters (leg length, step frequency, and body mass) closely match the experimental joint moments. However, there is some deviation during the double support phase. Although the

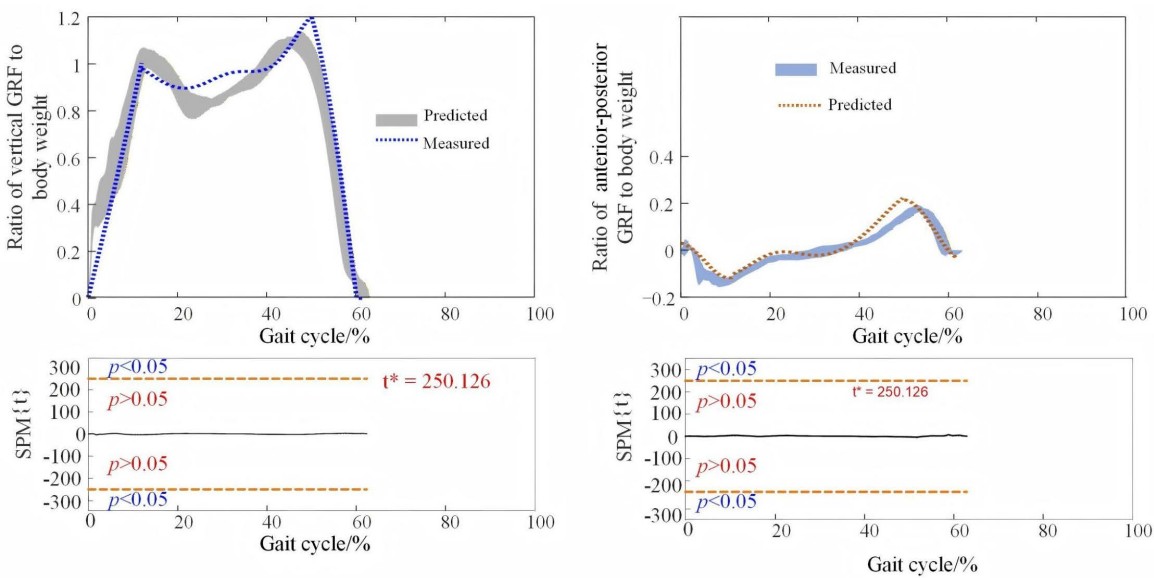

**Fig 5. The GRF across the entire gait cycle.** The dashed line denotes the model-predicted value. The shadow region is composed of 8 walking trials of one subject. The orange red dashed line indicates the critical thresholds (t* = 250.126, α = 0.05). Regions of the gait cycle for which SPM {t} exceeded the critical threshold are considered as the predicting model is invalid.

**Table 3. Error evaluation of 4 testing subjects by the GRF prediction model.**

| Testing subject | VGRF | | Anterior-posterior GRF | |
| --- | --- | --- | --- | --- |
| | rRMSE/% | $\rho$ | rRMSE/% | $\rho$ |
| 1 | 8.19 | 0.96 | 14.01 | 0.95 |
| 2 | 9.87 | 0.94 | 10.27 | 0.85 |
| 3 | 12.17 | 0.91 | 10.33 | 0.97 |
| 4 | 9.47 | 0.96 | 12.72 | 0.94 |
| Mean | 9.99 | 0.94 | 11.83 | 0.93 |

estimated hip and knee joint moments show an overall trend similar to that of the experimental data, there are discrepancies in specific details.

The predicted joint moments of the proposed model were also evaluated by rRSME and the correlation coefficient of the mean value of the measured 8 walking trials of each subject, and the errors are displayed in Table 4. The rRMSE ranged from 13.06% to 20.95%, and the correlation coefficient ranged from 0.84 to 0.93. Additionally, the errors of the ankle moments are relatively smaller than those of the hip and knee joints, and the errors of the knee joint moments are the largest. This is because the knee moment in the stance phase is derived directly from the hip moment and the ankle moment.

## 4. Discussions

This study proposed a predictive model of lower limb dynamics during walking using only leg length, body mass, and walking cadence, which are easy to measure. The proposed model is a personalized model with individuals' own physiological parameters. In real-world applications, leg length can be measured with a tape measure, body mass can be measured with a weight scale, and walking cadence can be measured with a second chronograph. With these easily

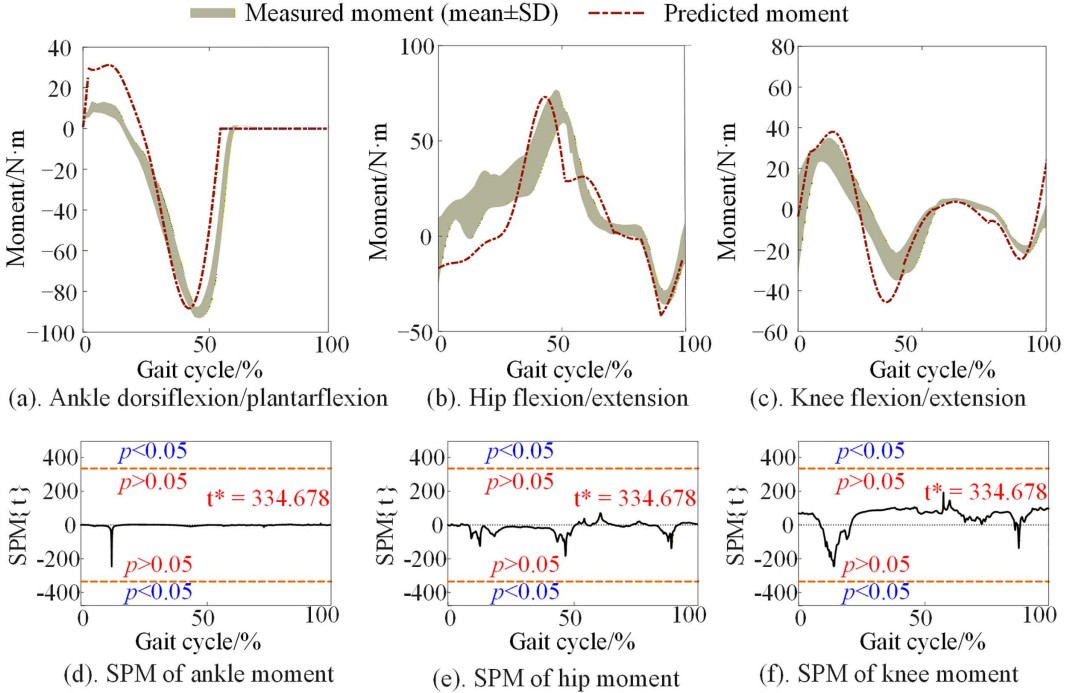

**Fig 6. The joint moment across the entire gait cycle of one subject.** The measurement moment in the figure is obtained from the walking trials of the corresponding subject in the experiment. The orange red dashed line indicates the critical thresholds (t* = 334.678, α = 0.05). Regions of the gait cycle for which SPM {t} exceeded the critical threshold are considered as the predicting model is invalid.

**Table 4. Error evaluation of the joint moment prediction model.**

| Testing subject | Ankle | | Hip | | Knee | |
|---|---|---|---|---|---|---|
| | rRMSE/% | ρ | rRMSE/% | ρ | rRMSE/% | ρ |
| 1 | 14.90 | 0.93 | 16.77 | 0.86 | 19.80 | 0.90 |
| 2 | 17.46 | 0.91 | 17.06 | 0.87 | 20.82 | 0.85 |
| 3 | 14.24 | 0.90 | 15.69 | 0.89 | 22.02 | 0.80 |
| 4 | 5.63 | 0.98 | 19.74 | 0.82 | 21.17 | 0.83 |
| mean | 13.06 | 0.93 | 17.32 | 0.86 | 20.95 | 0.84 |

measured parameters, the clinician can estimate the joint angle, moments, and ground reaction force for a patient when he/she is healthy. In addition, the proposed predictive model of the joint angle can also provide the customizer with his/her motion trajectory for the application of a human energy harvester [30].

Compared with similar studies, as shown in Table 5. The inertial measurement unit (IMU) is widely applied for the estimation of walking dynamics. The proposed methods can achieve superior performance for both the joint angle and moment, as well as the GRF with the simplest model and minimum sensors. In addition, the knee moment of these studies has a larger rRMSE than those of the hip and ankle [31–33,35–37]. Consistently, our proposed study also has the same results. In addition to healthy subjects, Ref [36] estimated the joint dynamics and GRF for poststroke patients.

There are several limitations of this study. The empirical parameters of the predictive model for the joint angle were obtained from 10 subjects, and the number of involved subjects was small, which may have resulted in inaccurate empirical parameters and led to larger errors. Additionally, the established predictive model assumes that the walking

**Table 5. Comparison of methods for estimating lower limb dynamics.**

| Ref | Required sensors | Models/ Methods | Joint angle | | Joint moment | | Anterior-posterior GRF | | Vertical GRF | |
|---|---|---|---|---|---|---|---|---|---|---|
| | | | rRMSE/% | ρ | rRMSE/% | ρ | rRMSE/% | ρ | rRMSE/% | ρ |
| Dorschky et al. 2019 [31] | 7 IMUs at pelvis, thighs, shanks, and feet | 7-segment musculoskeletal model | 8.8-21.9 | 0.96-1 | 14.4-27.1 | 0.76-0.95 | 9.7 | 0.95 | 9.6 | 0.95 |
| Lee et al. 2020 [32] | 1 IMU at sacrum | ANN | / | / | 11.4-24.08 | 0.20-0.77 | 9.21 | 0.88 | 6.7 | 0.93 |
| Li et al. 2021 [33] | 2 IMUs at shanks | 7-segment skeletal model | 11.5-26.1 | / | 20.4-29 | / | 3.9% | / | 12.2% | / |
| Ripic et al. 2022 [34] | A single azure kinetic sensor | musculoskeletal model | 6.7-16.4 | 0.488-0.876 | / | / | 4.1 | 0.889 | 2.9 | 0.94 |
| Hossain et al. 2023 [35] | 4 IMUs at thigh, shank, and foot 3 electronic goniometers at hip, knee and ankle | Kinetics-FM-DLR-Ensemble-Net | / | / | 5.59-6.06 | 0.765-0.868 | 3.77 | 0.922 | 5.49 | 0.957 |
| Simonetti et al. 2024 [36] | 5 IMUs at pelvis, thigh, shank and feet | EMG-driven musculoskeletal model | / | 0.51-0.66 | 32 | 0.67 | / | 0.46 | / | 0.86 |
| Liew et al. 2025 [37] | Walking speed measured by meter and second chronograph | Statistical model | 9.85-18.96 | 0.88-0.97 | 11.07-17.26 | 0.83-0.96 | 6.63 | 0.96 | 6.15 | 0.99 |
| This study | Walking cadence measured by second chronograph | 2 and3 -segment skeletal model | 7.38-9.84 | 0.97-0.99 | 13.06-20.95 | 0.84-0.93 | 11.83-12.93 | 0.92-0.93 | 9.99-11.05 | 0.93-0.94 |

characteristics across each gait cycle are periodically identical. While human walking is random and varies considerably, an individual cannot repeat two identical steps [38]. This discrepancy contributed to the errors between the predictive model and the experimental results. Generating a full time-series of estimates for each gait cycle will enable more robust validation of the model's performance across dynamic. The model's reliance on perfect symmetry and periodicity overlooks natural and clinically meaningful gait asymmetries, making it unsuitable for pathology assessment or rehabilitation monitoring, thus it is limited to approximating sagittal-plane biomechanics in healthy gait. In addition, relying on cadence as the sole temporal input oversimplifies gait, since cadence is influenced by factors such as speed, fatigue, and environment, limiting the model's diagnostic specificity. The model should therefore be viewed as a functional estimator of overall gait mechanics, with future work focusing on incorporating more specific biomarkers to improve clinical relevance.

The model generates an idealized, symmetric pattern, whereas therapy may aim to optimize safety, reduce pain, or improve energy efficiency, potentially by leveraging compensatory strategies that appear as asymmetries. Clinicians may use the model's output not as a goal but as a comparator. Future work will explore synergies with wearable technologies including advanced inertial measurement units [39] and electromyographic-driven models [40].

## 5. Conclusion

A dynamic model was developed to predict the sagittal plane joint angles, moments, and ground reaction forces during walking with a self-selected walking speed. The model requires only three inputs—leg length, body weight, and walking cadence—which can be measured with minimal instrumentation. To ensure continuity across the gait cycle, a linear transformation strategy was proposed to optimize the double support phase dynamics. Empirical kinematic formulas were first used to predict the hip and knee joint angles. These angle predictions were then integrated into the dynamic model through inverse dynamics to compute joint moments and GRFs. Validation against experimental data demonstrated consistency between the model predictions and the measured values. The results indicate that the proposed model can predict the joint angle, moment, and GRF in the sagittal plane. The proposed dynamic model demonstrates translational

potential for personalized rehabilitation and gait assistance devices, bridging computational innovation with clinical deployment.

## Acknowledgments

We sincerely acknowledge Ms. Pingping Wei from the School of Mechanical Engineering, Xi'an Jiaotong University, for her valuable technical assistance during the experimental setup and execution. We also extend our gratitude to all participants who contributed their time and effort to this study.

## Author contributions

**Conceptualization:** Guowu Wei, Qiumin Qu.

**Data curation:** Huan Zhao, Junxiao Xie, Anmin Liu.

**Formal analysis:** Huan Zhao.

**Funding acquisition:** Huan Zhao.

**Investigation:** Huan Zhao, Qiumin Qu.

**Methodology:** Huan Zhao, Wei-Hsin Liao.

**Software:** Huan Zhao.

**Supervision:** Junyi Cao, Wei-Hsin Liao.

**Validation:** Junyi Cao, Ziyun Ding.

**Visualization:** Anmin Liu, Ziyun Ding.

**Writing – original draft:** Huan Zhao.

**Writing – review & editing:** Junyi Cao, Ziyun Ding, Wei-Hsin Liao.

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
