## [Decision Letter · Decision Letter 0]

23 Jul 2025

Dear Dr. Cao,

Thank you for submitting your manuscript to PLOS ONE. After careful consideration, we feel that it has merit but does not fully meet PLOS ONE’s publication criteria as it currently stands. Therefore, we invite you to submit a revised version of the manuscript that addresses the points raised during the review process.

We look forward to receiving your revised manuscript.

Kind regards,

Masaya Anan, Ph.D.

Academic Editor

PLOS ONE

Journal Requirements: 

 [This work was supported by:

China Postdoctoral Science Foundation [Grant No. GZB20240603]

Author: [H.Zhao]

Funder: China Postdoctoral Science Foundation

URL: http://jj.chinapostdoctor.org.cn

National Key Research and Development Program of China [Grant No. 2021YFE0203400]

Author: [J.Cao]

Funder: Ministry of Science and Technology of the People's Republic of China

URL: http://www.most.gov.cn

Mainland-Hong Kong Joint Funding Scheme (MHKJFS) [Project No. MHP/043/20]

Author: [W.H. Liao]

Funder: Innovation and Technology Commission, Hong Kong Special Administrative Region

URL: https://www.itc.gov.hk

The Chinese University of Hong Kong [Project ID: 3130174]

Author: [H.Zhao]

Funder: The Chinese University of Hong Kong

URL: https://www.cuhk.edu.hk

Direct Grant for Research 2024/2025 [Project ID: 4055254]

Author: [H.Zhao]

Funder: The Chinese University of Hong Kong

URL: https://www.cuhk.edu.hk]. 

Reviewers' comments:

Reviewer's Responses to Questions

**Comments to the Author**

1. Is the manuscript technically sound, and do the data support the conclusions?

Reviewer #1: Yes

Reviewer #2: Yes

2. Has the statistical analysis been performed appropriately and rigorously?

Reviewer #1: Yes

Reviewer #2: Yes

3. Have the authors made all data underlying the findings in their manuscript fully available?

Reviewer #1: Yes

Reviewer #2: No

4. Is the manuscript presented in an intelligible fashion and written in standard English?

Reviewer #1: Yes

Reviewer #2: Yes

Reviewer #1: I consider that the study is performed in accordance with requirements. The methodology is properly explained. The study claims it is an easy to measure technique.

I specifically appreciated the high applicability of the described method.

Reviewer #2: 1. Overall Evaluation

This manuscript presents a predictive model for estimating sagittal-plane gait dynamics (joint angles, joint moments, and ground reaction forces) based on only three input variables: leg length, body mass, and cadence. The minimal data requirements make this model potentially attractive for use in low-resource clinical or field settings. The mathematical structure is well developed and the model is validated against experimental data from healthy participants.

However, several conceptual and practical limitations significantly affect the model’s generalizability and translational value. These limitations are either insufficiently discussed or entirely unacknowledged in the manuscript. The authors are encouraged to clarify the intended scope of the model and to openly address its current boundaries, particularly regarding its clinical utility.

2. Strengths

- The model is simple, uses non-invasive and easily obtainable inputs, and has potential for low-cost deployment.

- It captures the full gait cycle, including both single and double support phases.

- The mathematical derivations are clearly presented and fully reproducible in principle.

- The inclusion of mass transfer during the double support phase is a novel and elegant feature for ensuring continuity of force and moment trajectories.

- Comparative performance is presented relative to existing models in the literature.

3. Major Limitations

3.1. Assumption of gait symmetry and cycle periodicity

The model assumes perfect left–right symmetry and step-to-step periodicity. While the authors briefly acknowledge that human gait is inherently variable, they do not fully discuss the practical implications of this assumption. Gait asymmetry and temporal variability are not only normal but often clinically informative—especially in populations with stroke, Parkinson’s disease, or orthopedic impairments. The assumption of symmetry limits the model’s applicability to pathological gait or real-world functional assessments. This point should be discussed in terms of how it restricts the model’s diagnostic and rehabilitative utility.

3.2. Use of cadence as the primary dynamic input

The model relies on cadence as the only time-dependent input variable. Although cadence is easy to measure, it is not a specific or sensitive indicator of gait pathology. Cadence changes may reflect alterations in walking speed, fatigue, or motivation, but not necessarily impairment. The manuscript does not address this limitation, nor does it review the limited clinical value of cadence alone as a predictor of gait quality. This is a significant conceptual gap that should be acknowledged and discussed.

In addition, the manuscript does not address how cadence should be measured outside of laboratory environments. To ensure ecological applicability, the authors should recommend validated mobile or wearable tools, define acceptable accuracy thresholds, and consider performing a validation comparing cadence from reference systems (e.g., VICON) to field-based methods (e.g., inertial sensors or apps). Without this, the practical deployment of the model remains limited.

3.3. Use of a theoretical or ideal gait reference

The model appears to be designed as a generator of an idealized gait profile, which could serve as a target or reference for a given individual based on anthropometry and cadence. However, in clinical rehabilitation, the goal is not to restore a normative gait pattern but to optimize functionality, safety, and energy efficiency, which may involve accepting asymmetry or compensation. The authors suggest this use case but do not critically examine its limitations. The potential mismatch between theoretical predictions and real-world therapeutic goals should be addressed.

3.4. Lack of accessible implementation contradicts the model’s purpose

Although the model is mathematically well-documented and could be implemented using open-source environments such as R or Python, the authors do not provide any source code or mention future code availability. This omission limits reproducibility and transparency. More importantly, it contradicts the model’s stated goal of providing a simple and accessible alternative to expensive lab-based motion analysis. If the method is to be truly usable in low-resource settings, the implementation must also be openly available.

The authors should share the source code in a public repository (e.g., GitHub, Zenodo, OSF) using an open-access license and a programming language compatible with widespread use. This would allow researchers and clinicians to reproduce the results, adapt the model to specific populations, and ensure consistency in future validations.

4. Formatting and Presentation

- Figures: Figures are clearly referenced and support the validation process effectively. However, captions should be expanded to clarify curve identities, sample size, and units.

- Tables: Table 5 is difficult to interpret due to formatting complexity. It should be restructured for clarity, possibly in landscape format or as a supplementary file.

- Symbols and notation: A glossary or table of variables would improve clarity and help readers follow the mathematical models more easily.

**Do you want your identity to be public for this peer review?** For information about this choice, including consent withdrawal, please see our Privacy Policy

Reviewer #1: **Yes: ** Diana Ciubotariu

Reviewer #2: No

---

## [Author Response · Author response to Decision Letter 1]

5 Sep 2025

Respond to editor comments and reviewers are listed in the "Response Letter to Reviewers".

---

## [Decision Letter · Decision Letter 1]

24 Sep 2025

Dear Dr. Cao,

Thank you for submitting your manuscript to PLOS ONE. After careful consideration, we feel that it has merit but does not fully meet PLOS ONE’s publication criteria as it currently stands. Therefore, we invite you to submit a revised version of the manuscript that addresses the points raised during the review process.

We look forward to receiving your revised manuscript.

Kind regards,

Andrea Tigrini, Ph.D.

Academic Editor

PLOS ONE

Journal Requirements:

**Additional Editor Comments:**

The paper has been revised by three experts in the field. The manuscript has been recognied good by all the reviewers however some clarifications are needed. Please respond to the reviewers concerns point-by-point.

Reviewers' comments:

Reviewer's Responses to Questions

**Comments to the Author**

Reviewer #1: All comments have been addressed

Reviewer #2: All comments have been addressed

Reviewer #3: (No Response)

2. Is the manuscript technically sound, and do the data support the conclusions?

Reviewer #1: Yes

Reviewer #2: Yes

Reviewer #3: Yes

3. Has the statistical analysis been performed appropriately and rigorously?

Reviewer #1: (No Response)

Reviewer #2: Yes

Reviewer #3: N/A

4. Have the authors made all data underlying the findings in their manuscript fully available?

Reviewer #1: (No Response)

Reviewer #2: Yes

Reviewer #3: No

5. Is the manuscript presented in an intelligible fashion and written in standard English?

Reviewer #1: (No Response)

Reviewer #2: Yes

Reviewer #3: Yes

Reviewer #1: The maniscript is in good shape for publishing. The questions of the reviwers were properly answered.

Reviewer #2: I would like to thank the authors for the detailed responses to my comments. I think the manuscript is now suitable for publication. Congrats!

Reviewer #3: The study of Zhao and colleagues aims to develop a minimum input model for joint angle and GRF estimation based on a bipedal mechanism that switches between two subsystems, i.e., one related to the double support phase and one to the single support phase. Overall, the research topic is valid; however, many concerns are present in the paper, and I list below my main points.

MAIN ISSUES

1. In the methods section, for the Fourier approximation, the authors must specify how many harmonics they considered in the model and how they tuned the coefficients. Although this can later be inferred from Table 10, it should be clearly reported in the methods section.

2. A major problem concerns the correspondence between the equations and Figures 2 and 3. The terms used in the equations are not clearly indicated in the figures, which makes the manuscript confusing and difficult to interpret. For instance, in Equation 3, the authors refer to the double support figure (Fig. 3), but they use the terminology Iand i-th, which seems more appropriate for Figure 2. Moreover, Equation 3 shows only two segments. This is just one example, but in general, there is a lack of consistency between the quantities shown in the figures and those indicated in the equations (consider also the masses of the segments). As a result, it is difficult to verify and check the equations against the figures.

3. The authors used different cycles to estimate the cadence frequency. This is why the estimate results in one gait cycle, while in the data there may be multiple gait cycles. Although this can be understood, it is not properly reported in the methods section and should be clarified. The estimated data essentially represent an average. Instead, the authors should work on producing a time series of the trace according to the model for each gait cycle.

4. The metrics computed are good but not sufficient. After being able to estimate the data, the authors should also use Statistical Parametric Mapping (SPM), which is freely available as software packages in both MATLAB and Python.

5. In the comparison table, the authors should expand the review to include methods based on EMG and IMU signals, which are wearable and represent a good framework for estimation. From this perspective, they could revise their work by considering the following manuscripts:

• Scattolini, Mara, et al. "Inertial Sensing for Human Motion Analysis: Enabling Sensor-to-Body Calibration Through an Anatomical and Functional Combined Approach." IEEE Transactions on Neural Systems and Rehabilitation Engineering (2025).

• Mengarelli, Alessandro, et al. "Myoelectric-Based Estimation of Vertical Ground Reaction Force during Unconstrained Walking by a Stacked One-Dimensional Convolutional Long Short-Term Memory Model." Sensors24.23 (2024): 7768.

**Do you want your identity to be public for this peer review?** For information about this choice, including consent withdrawal, please see our Privacy Policy

Reviewer #1: **Yes: ** Diana Ciubotariu

Reviewer #2: **Yes: ** Eduardo Carballeira

Reviewer #3: No

---

## [Author Response · Author response to Decision Letter 2]

9 Nov 2025

Respond to reviewers is attached in the file.

---

## [Editor Report · Decision Letter 2]

18 Nov 2025

A predictive model of joint dynamics and ground reaction force using only leg length, body mass, and walking cadence

PONE-D-25-28685R2

Dear Dr. Cao,

We’re pleased to inform you that your manuscript has been judged scientifically suitable for publication and will be formally accepted for publication once it meets all outstanding technical requirements.

Kind regards,

Andrea Tigrini, Ph.D.

Academic Editor

PLOS ONE
---

## [Editor Report · Acceptance letter]

PONE-D-25-28685R2

PLOS One

Dear Dr. Cao,

I'm pleased to inform you that your manuscript has been deemed suitable for publication in PLOS One. Congratulations! Your manuscript is now being handed over to our production team.

Kind regards,

on behalf of

Dr. Andrea Tigrini

Academic Editor

PLOS One